# Characterization of PRDM9 Multifunctionality in Yak Testes Through Protein Interaction Mapping

**DOI:** 10.3390/ijms26041420

**Published:** 2025-02-08

**Authors:** Guowen Wang, Shi Shu, Changqi Fu, Rong Huang, Shangrong Xu, Jun Zhang, Wei Peng

**Affiliations:** 1State Key Laboratory of Plateau Ecology and Agriculture, Qinghai University, Xining 810016, China; wangguowen@qhu.edu.cn; 2Qinghai Academy of Animal Science and Veterinary Medicine, Qinghai University, Xining 810016, China; shushi0211@163.com (S.S.); fuchangqimau@163.com (C.F.); mkyhuangr@163.com (R.H.); 2008.xushangrong@163.com (S.X.); 3Northwest Institute of Plateau Biology, Chinese Academy of Sciences, Xining 810008, China; 4Qinghai Provincial Key Laboratory of Pathogen Diagnosis for Animal Diseases and Green Technical Research for Prevention and Control, Xining 810016, China

**Keywords:** PRDM9, yak, protein interaction, meiotic recombination, hybrid sterility

## Abstract

Meiotic recombination is initiated by the formation of programmed DNA double-strand breaks during spermatogenesis. PRDM9 determines the localization of recombination hotspots by interacting with several protein complexes in mammals. The function of PRDM9 is not well understood during spermatogenesis in mice or yaks. In this study, we applied yeast two-hybrid assays combined with next-generation sequencing techniques to screen the complete set of PRDM9-interacting proteins and explore its novel functions in yak spermatogenesis. Our results showed that 267 PRDM9-interacting proteins were identified. The gene ontology (GO) analysis of the interacting proteins revealed that the GO terms were primarily associated with spermatogenesis, positive regulation of double-strand break repair via homologous recombination, RNA splicing, the ubiquitin-dependent ERAD pathway, and other biological processes. MKX and PDCD5 were verified to be strongly interacting with PRDM9 and expressed in prophase I of meiosis in both mouse and yak testes. The localizations of RNA splicing genes including *THOC5*, *DDX5*, and *XRCC6* were expressed in spermatocytes. Cattleyak is the hybrid offspring of a yak and a domestic cow, and the male offspring are sterile. The gene expression of the interacting proteins was also examined in the sterile male hybrid of yak and cattle. Among the 58 detected genes, 55 were downregulated in cattleyak. In conclusion, we established a complete PRDM9 interaction network, and a novel function of PRDM9 was identified, which will further promote our understanding of spermatogenesis. It also provides new insights for the study of hybrid male sterility.

## 1. Introduction

Spermatogenesis is a highly complicated, specialized, and tightly regulated cell developmental process, including the self-renewal and differentiation of spermatogonia, meiosis of spermatocyte, and spermiogenesis [1]. In mammals, sustainable spermatogenesis relies on the replenishment of the spermatogonial stem cell (SSC) pool and their differentiation. Retinoic acid is responsible for the spermatogonial A-to-A1 transition and meiotic entry and simultaneously induces the expression of Stimulated By Retinoic Acid 8 (STRA8), which is essential for the initiation of meiosis [2]. Meiotic recombination is a key event during meiosis that occurs from the leptotene stage to the pachytene stage [3]. Meiotic recombination assures accurate segregation of homologous chromosomes and enhances the genetic diversity. It is initiated by the induction of programmed DNA double-strand breaks (DSBs) at preferred genome loci at meiosis, called hotspots, which control the frequency distribution of recombination [4]. In mice and humans, DSB hotspots are determined by PRDM9, a DNA sequence-specific binding histone methyltransferase which generates trimethylation on lysine 4 (H3K4me3) and lysine 36 (H3K36me3) of histone H3 [5]. PRDM9 is a key regulator of meiotic recombination and mainly expressed in spermatocytes. It is composed of several functional domains, including the N-terminal region, which contains KRAB/SSXRD domains that mediate protein interactions; the central region, which comprises the PR/SET domain that is responsible for methyltransferase activity; and the C-terminal region, which consists of tandem zinc finger domains that participate in DNA binding. The zinc finger domain that is located at the C-terminal region is highly polymorphic and represents one of the most rapidly evolving regions in the genome.

PRDM9 regulates the localization of recombination hotspots by interacting with protein complexes and establishing the local epigenetic environment. HELLS, a chromatin remodeling enzyme, interacts with PRDM9, forming a pioneer complex to facilitate 5-hydroxymethylcytosine (5hmC) deposition and the formation of open chromatin at meiotic recombination hot spots [6,7]. Subsequently, CXXC1, EWSR1, EHMT2, and CDYL directly interact with PRDM9, as well as with cohesin REC8 and the synaptonemal complex proteins SYCP3 and SYCP1, to create a spatial environment that facilitates the induction of DSB formation [8,9,10]. Recent research showed that ZCWPW1 coevolved with PRDM9. As a histone modification reader, ZCWPW1 can recognize H3K4me3 and H3K36me3 histone modifications, further involving it in homology search and DSB repair [11,12]. However, PRDM9 and its binding sites are not sufficient for DSB formation; additional factors that are involved in DSB formation still need to be identified.

*PRDM9* is the only speciation gene that has been identified in vertebrates. Differences in its zinc finger domain between species result in altered binding properties, ultimately leading to hybrid male sterility [13]. However, it is currently uncertain whether this phenomenon is applicable to other hybrid species. Cattleyak, the hybrids of cattle (*Bos taurus*) and yak (*Bos grunniens*), are widely produced due to heterosis. The male hybrid is completely sterile due to the reduced proliferation of undifferentiated spermatogonia and the impairment of meiotic recombination [14,15]. The localization of meiotic recombination hotspots is PRDM9-dependent in bovine [16]. However, it is still unclear whether *PRDM9* is the cause of sterility in cattleyak.

In this study, we applied yeast two-hybrid assay sequencing (Y2H-seq) to identify novel PRDM9-interacting proteins in yak testes. Our results indicate that PRDM9 interacts with multiple proteins in vitro and might be involved in different biological processes. PRDM9 and its interactive proteins may be responsible for male sterility in cattleyak.

## 2. Results

### 2.1. PRDM9 Expression Pattern in Yak and Cattleyak Spermatocytes

DSB hotspot localization in bovids is PRDM9-dependent. Our previous results showed that the PRDM9 expression in adult yak testes was significantly higher compared to in adult cattleyak [17]. In this part, we detected the localization of PRDM9 in yak and cattleyak testes. The adult yak seminiferous epithelium consists of all types of germ cells, including spermatogonia, spermatocytes, and spermatids. The Sertoli cells are located at the basement membrane of the seminiferous tubules (Figure 1A). The histomorphology of the cattleyak testis clearly showed defects in spermatogenic cells, with only spermatogonia and spermatocytes being observed (Figure 1B). γH2AX is a putative marker of leptotene/zygotene and pachytene spermatocytes, which undergo dynamic changes during prophase of meiosis I. Our results showed that PRDM9 and γH2Ax were co-stained in the testes of yak and cattleyak. In yak, leptotene/zygotene and pachytene spermatocytes were distinguished based on the γH2AX-positive signal. In pachytene spermatocytes, the γH2AX-positive signals were focused on the sex body. PRDM9 was localized in the leptotene/zygotene stage of the spermatocytes, and no positive signals were detected for pachytene spermatocytes in yak. In the cattleyak testis, a γH2Ax-positive signal was detected; however, no sex body was observed. PRDM9 was also co-localized in γH2Ax-positive cells in cattleyak. These results indicate that PRDM9 localization shows the same pattern in both yak and cattleyak.

### 2.2. Self-Activation Test

To verify whether a self-activation phenomenon exists, the plasmids pGBKT7-PRDM9 and pGBKT7 were transformed into the AH109 yeast strain and cultured on SD/-Trp plates. Subsequently, six positive colonies were randomly selected, and PCR verification was performed using the pGBKT7 vector primers, all of which showed correct cloning. Three additional randomly selected colonies were then transferred to SD/-Trp, SD/-Trp/-His, SD/-Trp/-His/-Ade, and SD/-Trp/-His/-Ade+X-α-gal plates. The pGBKT7 vector served as a negative control. The results showed that the negative control, pGBKT7, could grow on SD/-Trp plates but could not grow normally on SD/-Trp/-His, SD/-Trp/-His/-Ade, or SD/-Trp/-His/-Ade+X-α-gal plates (Figure 2). The growth of the experimental group was consistent with that of the control group.

### 2.3. A Yeast Two-Hybrid Assay Identifies Multiple PRDM9 Interactors in the Yak Testes

The determination of DSB hotspot is regulated by PRDM9 and its interaction network; however, the network remains to be completed. To search for global proteins that are directly interacting with PRDM9 in yak testes, we applied the Y2H system to screen a cDNA library from different adult yak testes. At first, we performed a Y2H assay using cloned full-length PRDM9 as bait and a cDNA library derived from three mixed adult yak testis tissues as prey. Based on the Y2H assay, 192 positive yeast clones, which grew on synthetic dextrose medium-lacking tryptophan, leucine, and histidine (SD-TLH) plates, were selected for PCR and sanger sequencing. Through sequence alignment, we identified 43 gene sequences (Appendix A). Subsequently, we performed yeast-positive clone re-transformation verification by seeding the positive clones that grew on SD-TLH-deficient plates onto a synthetic dextrose medium lacking leucine and histidine (SD-TL); SD-TLH; a synthetic dextrose medium lacking tryptophan, leucine, histidine, and adenine (SD-TLHA); and SD-TLHA+X-α-gal-deficient plates. The positive control can grow on SD-TL, SD-TLH, SD-TLHA, and SD-TLHA+X-α-gal-deficient plates and appears blue on the SD-TLHA+X-α-gal-deficient plate. The negative control can only grow on the SD-TL plate and cannot grow on the other plates. The positive clones that were grown on SD-TLHA-deficient plates were proven to be PSMA6, ELP5, CHD4, TMBIM4, EDF1, and SCNM1. Next, we combined the Y2H assay with next-generation sequencing to detect the PRDM9 interactome at the whole-transcriptome scale. All positive clones that grew on SD-TLH-deficient plates were collected in a yeast extract peptone dextrose (YPD) medium for PCR, and the resulting PCR products were used for next-generation sequencing. The results identified 267 PRDM9-interacting proteins, including 3 transcription factors (LOC616431, ZNF24, MKX) and 16 kinases (ROCK1, DAPK3, IFT27, STK16, MAP2K7, TNIK, NDUFA10, TESK2, SRPK1, NEK2, HSP90AA1, HSPA9, TTBK2, CDKN3) (Appendix A). All 43 proteins that were identified in the Y2H assay were detected in the Y2H-seq data. REC114 and FUS proteins were identified, with read counts of 1068 and 580, respectively, and they were proven to interact with PRDM9 and played a role in promoting the initiation of meiotic recombination in mouse testes. The top 20 PRDM9-interacting proteins are listed in Table 1, and their locations are discussed below.

To further analyze the functions of PRDM9-interacting proteins, we conducted a Kyoto Encyclopedia of Genes and Genomes (KEGG) analysis and gene ontology (GO) analysis. The pathways were mainly enriched in protein processing in the endoplasmic reticulum, ubiquitin-mediated proteolysis, ribosome, and proteasome (Figure 3A). The GO terms were mainly focused on spermatogenesis (DNAJA1, SBF1, CALR3, IFT27, MEA1, ODF2, PRM2, SNRPA1, SLC25A31, SPATA32, SYCE3, TDRD1), positive regulation of double-strand break repair via homologous recombination (FUS, MORF4L1, and ARID2, with read counts of 580, 10,582, and 6, respectively), RNA splicing (THOC5, SCNM1, DDX5, XRCC6), protein folding, positive regulation of telomere maintenance via telomerase, protein stabilization, ubiquitin-dependent ERAD pathway, and other biological processes, which indicates that PRDM9 might be involved in multiple biological processes (Figure 3B). 

### 2.4. Localization of Screened PRDM9-Interacting Proteins in Testes of Mouse and Cattleyak

To further verify the Y2H-seq data, we applied a HDOCK server to predict the crystal structure for protein interaction and the interaction intensity strength. Mohawk homeobox (MKX) and programmed cell death 5 (PDCD5)-ranked top reads were selected to analyze their interaction with PRDM9 using the HDOCK server. We submitted the complete amino acid sequences to the HDOCK server for the analysis of the interaction intensity strength [18]. Our results showed that the crystal structures were tightly interacting and that the confidence scores for PRDM9 interactions with MKX and PDCD5 were 0.9452 and 0.9327, respectively (Figure 4), indicating their high interaction intensity strength. Next, we co-stained MKX and PDCD5 with γH2AX, as well as SYCP3, a putative marker of the synaptonemal complex, to detect the localization of the two proteins in adult mouse, yak, and cattleyak testes. We observed that both proteins were co-expressed in SYCP3-positive cells in adult mouse testes and in γH2AX-positive cells in yak and cattleyak testes. However, they exhibited different expression patterns. MKX was localized in leptonema at the basement membrane and in the sex body of pachynema in mice. However, in yak and cattleyak testes, MKX was also co-expressed with γH2AX-positive cells but was diminished in the sex body during pachynema (Figure 5). The strong staining of interstitial cells in Figure 5A–C is non-specific staining. The co-staining results of PDCD5 and γH2Ax showed that PDCD5 was localized in spermatocytes from the zygotene to the pachytene of mice, yak, and cattleyak. Sertoli cells were also positive for PDCD5 in all three animals (Figure 6).

### 2.5. The Expression Profiles of PRDM9-Interacting Protein-Coding Genes in Cattleyak

As mentioned above, male cattleyak are completely sterile. Our previous results reported that the expression of *PRDM9* in cattleyak is significantly decreased compared to in adult yak [17]. The zinc finger array sequence of yak is quite different from that of cattle, which might explain the male sterility observed in their hybrids, although this has not yet been verified [19]. To further reveal its role in cattleyak spermatogenesis, we conducted a correlation analysis between previously published transcriptome data in yak and cattleyak and the Y2H-seq data in this study. According to our analysis, 58 genes that were identified in Y2H-seq were significantly differentially expressed, with 55 genes bein downregulated in cattleyak compared to yak (Appendix A). A GO analysis revealed that the 58 genes were enriched in spermatogenesis, regulation of the attachment of spindle microtubules to kinetochore, protein folding, nuclear envelope organization, response to unfolded protein, centriole replication, positive regulation of the apoptotic process, cell division, and Rho protein signal transduction (Appendix A). These genes may cooperate with PRDM9 to regulate spermatogenesis in yak testes.

### 2.6. Construction of Protein–Protein Interaction Networks of PRDM9 in Yak Testes

According to the Y2H-seq, we identified a batch of proteins that directly interact with PRDM9 in yak testes. Here, by combining published data from mice and Y2H-seq data on PRDM9 from this study, we constructed a complete PRDM9 interaction network (Figure 7). So far, 14 proteins have been found to directly interact with PRDM9 and function in DSB formation and homology search. Many of these genes were identified in our study and not previously associated with DSB processing or spermatogenesis and may represent unrecognized components of the recombination machinery. The novel functions of PRDM9 in spermatogenesis need further validation.

## 3. Discussion

PRDM9 plays an indispensable role of orchestrating crucial stages of meiotic recombination initiation in most mammals. The function of PRDM9 in the localization of DSB hotspots and DSB repair during meiotic recombination requires the involvement of multiple proteins. These proteins collaborate to ensure the precise targeting and activation of DSB sites. In this study, we detected the localization of PRDM9 in both yak and cattleyak testes and employed Y2H-seq to identify the global PRDM9-interacting proteins, allowing us to establish a comprehensive PRDM9 interaction network and reveal new functions of PRDM9 and discover novel proteins that are involved in DSB processing during spermatogenesis in yak testes.

PRDM9, a histone lysine methyltransferase, is expressed in the mouse testis, where it initially appears at preleptotene and persists until the late zygotene stage [20]. In this study, we performed co-staining of γH2AX, a DSB marker that can distinguish between leptotene/zygotene and pachytene during meiosis, with PRDM9 in the testes of yak and its hybrid progeny, cattleyak. The sex body was observed in yak; however, no sex body was detected in cattleyak, indicating that the meiotic progression of cattleyak was interrupted. The immunofluorescence staining results showed that PRDM9-positive signals were co-localized with γH2AX, except for the pachytene stage marked by γH2AX in yak and cattleyak testes. For its function in DSB repair, PRDM9 might be retained on chromosomes in pachytene-like cells.

Y2H-seq is a high-efficiency and reliable method to identify interacting proteins of a known protein [21]. Previous studies have reported that PRDM9 interacts with various proteins and functions in meiotic recombination; however, it is still unclear whether there are any proteins that interact with PRDM9. In this study, Y2H-seq was employed as a powerful technique to systematically identify interacting proteins of PRDM9, enabling the construction of an interaction network that uncovers both well-established and novel protein partners. By screening PRDM9 against a comprehensive cDNA library, to our surprise, we detected 267 proteins that interact with PRDM9. The GO analysis of the PRDM9-interacting proteins indicate that PRDM9 not only plays a significant role in the orchestration of meiotic recombination but is also involved in other biological processes that are associated with the regulation of spermatogenesis. To our surprise, MORF4L1 and ARID2 were found for the first time to interact with PRDM9 and might function in double-strand break repair in yak. MORF4L interacts with the entire BRCA complex, which contains BRCA1, PALB2, BRCA2, and RAD51 [22], even though the two proteins were detected in pachytene spermatocytes in mouse [23,24]. RNA splicing is also a key factor for the regulation of spermatogenesis. In addition to THOC5, SCNM1, DDX5, and XRCC6, which are mentioned above, we have also identified the functions of SRSF3, SFSWAP, SBF1, SNRPA, and CCDC12 in RNA splicing. During spermatogenesis, it is important to maintain the ubiquitin level in germ cells. In the Y2H-seq data, the following ubiquitin-related genes were detected: CALR3, HSP90B1, SGTA, UBE2G2, UBA52, UBB, UBE2G2, UBE2H, UBE2V1, UPF2, USP4, USP47, and USPL1. We may infer that timely degradation of PRDM9, which depends on the ubiquitin pathway, is essential.

AI models, such as AlphaFold (AlphaFold-Multimer predicts cross-kingdom interactions at the plant–pathogen interface), HDOCK [25], and RFdiffusion [26], are becoming popular for predicting protein–protein interactions. We also predicted the interactions of MKX and PDCD5 with PRDM9 using the HDOCK server and confirmed that they have strong interactions. Both proteins are localized in the spermatocytes of mouse and yak and its hybrid. MKX was expressed in spermatocytes from the leptotene stage to the pachytene stage and were partially co-localized with PRDM9. As previously reported, the PRDM9 expression ranges from preleptotene and leptotene to mid-zygotene spermatocytes [20]. MKX is a potent transcriptional repressor that is expressed in the embryonic precursors of skeletal muscle [27], and it is crucial for the repair of adult skeletal muscle [28]; however, as a transcriptional repressor, its role in spermatogenesis is still not known. PDCD5 functions in multiple cellular processes, including apoptosis and autophagy; it facilitates p53-dependent apoptosis through its interaction with mouse double minute 2 homolog (MDM2) [29]. It is the first time that PDCD5 expression in mouse spermatocytes has been reported. During meiotic recombination in spermatogenesis, programmed DSBs are formed. P53 is responsible for DNA damage repair [30], and PDCD5 might be involved in this process. In yak and cattleyak, the PDCD5 expression started from zygotene spermatocytes and ended at the pachytene stage. In cattleyak, the expression of PDCD5 seems to be lower compared to yak. A previous report showed that the number of apoptotic cells is significantly higher in cattleyak than in yak, which suggests that other pathways might be involved in the apoptosis of germ cells in cattleyak [31]. PDCD5 was also detected in Sertoli cells in mice, yak, and cattleyak. So far, the conditional knockout of PDCD5 in both germ cells and Sertoli cells has not been reported. The role of PDCD5 in spermatogenesis requires further exploration.

RNA splicing has been proven to be an important event during spermatogenesis; however, its function remains largely unknown [32]. In this research, the RNA-splicing factors THOC5, SCNM1, DDX5, and XRCC6 were found to interact with PRDM9. A GO analysis showed that the PRDM9-interacting proteins were enriched in RNA splicing, providing a new role of PRDM9 in spermatogenesis. DDX5 (also called P68) and XRCC6 (also called KU70), which relate to RNA splicing, were reported to be expressed in mouse testes. DDX5 has been reported to be expressed in all spermatogenic cells, from spermatogonia to spermatids, and regulates the splicing of key genes that are necessary for spermatogenesis [33]. XRCC6 is mainly expressed in Sertoli cells, spermatogonia, and late spermatocytes and is important for repairing double-strand breaks (DSBs) in late spermatocytes in the testis [34,35]. THOC5, SCNM1, SRSF3, SFSWAP, SBF1, SNRPA, and CCDC12 are also important in RNA splicing; however, how these proteins interact with PRDM9 to further exert their function remains to be studied.

The cattleyak serves as a valuable model for investigating hybrid male sterility, which enhances our understanding of speciation. As a speciation gene, PRDM9 primarily functions in DSB hotspot selection. However, its role in yak and its hybrids remains poorly studied. Whether the downregulation of PRDM9 and its interactors in cattleyak or the diversity of its zinc finger array leads to meiotic arrest is still unclear. Combined with transcriptome data, 55 PRDM9-interacting proteins that were identified in this study were significantly downregulated in cattleyak compared to yak. CETN1, CRISP2, and BTG4 were rarely expressed in cattleyak testes, suggesting that the downregulation of PRDM9 interactors may contribute to meiotic arrest in cattleyak.

## 4. Materials and Methods

### 4.1. Animals and Sample Preparation

All animal experimentation was performed in accordance with institutional ethical guidelines and were approved by the Qinghai University Animal Ethics Committee. Testes were collected from six wild-type C57BL6N mice at the age of 60 days immediately after sacrifice by cervical dislocation and then embedded in paraffin according to standard procedures for cross-sectioning. Testes from six 36-month-old healthy yaks and cattleyaks, respectively, were obtained from a licensed commercial abattoir (Qinghai Xiahua Livestock Products Co., Ltd., Haibei, Qinghai Province, China) in July. Upon collection, the epididymis and tunica albuginea were removed, and the tissues were processed as follows: one portion of the testis was fixed in 4% paraformaldehyde at room temperature and then paraffin-embedded for histological analysis, while the other portion was snap-frozen in liquid nitrogen and stored at −80 °C for subsequent RNA extraction.

### 4.2. Yeast Two-Hybrid (Y2H) and Y2H-seq Assay

The Y2H assay was performed as previously described [21]. Briefly, yak (*Bos grunniens*) *PRDM9* CDS (accession number: KJ020104.2) was cloned into the pGBKT7 (BD). We introduced recombinant plasmids into the yeast strain through the yeast two-hybrid (Y2H) system (Ruiyuan Biotechnology Co. Ltd. (Nanjing, China)). Transformants were plated on an SD (synthetic dropout) -Leu-Trp medium and cultured at 30 °C for 2 days. Interaction assays were performed on SD Trp-Leu-His-Ade plates, followed by incubation at 30 °C for 3–4 days. For the Y2H-seq assay, we prepared the yeast library using RYbiotech company tools (Ruiyuan Biotechnology Co. Ltd. (Nanjing, China)). Total RNA was isolated from three different yak testes, followed by reverse transcription to generate the complete cDNA from the yak testes. Subsequently, the whole cDNA was inserted into pGADT7 vectors (AD). Finally, the expression vectors were transformed into the AH109 strain to generate a working liquid. Meanwhile, pGBKT7-PRDM9 plasmids were used to test for self-activation and screen for PRDM9-interacting proteins using the Y2H mating protocol (RYbiotech company). All positive clones from the yeast library screening plates were collected into a liquid medium. After centrifugation to collect the yeast cells, colony PCR was performed, followed by NGS sequencing of the PCR products.

### 4.3. Yeast Two-Hybrid Screen Validation and Prdm9 Domain Mapping

The positive clones that grew on the SD-TLH dropout plates were individually diluted with sterile water and spotted onto SD-TL, SD-TLH, SD-TLHA, and SD-TLHA+X-α-gal dropout plates. The plates were then incubated at 30 °C for 3–4 days. The positive control grew on SD-TL, SD-TLH, SD-TLHA, and SD-TLHA+X-α-gal plates and displayed a blue color on the SD-TLHA+X-α-gal plates. The negative control only grew on the SD-TL plates and did not grow on any of the other plates.

### 4.4. Gene Ontology (GO) and KEGG Analysis

The DAVID database (https://david.ncifcrf.gov/ (accessed on 4 November 2024)) was used for functional annotation [36]. The gene list was uploaded to DAVID to generate functional clusters and KEGG pathways.

### 4.5. Histomorphology and Immunofluorescence Staining [9]

Testicular tissues fixed in 4% PFA were applied to conduct hematoxylin and eosin (H&E) staining and immunohistochemical staining. Briefly, the tissues were embedded in paraffin wax and sectioned at 5 µm. For the histomorphology, the paraffin sections were deparaffinized with xylene and rehydrated with a gradient of alcohol, followed by H&E staining. For the immunohistochemical staining, the rehydrated slides were boiled in 10 mM trisodium citrate buffer (pH 6.0) in a microwave oven for 20 min for antigen retrieval, then washed with PBS, and further blocked with 10% normal donkey serum in PBS for 1 h at room temperature. The slides were then incubated with primary antibodies that had been diluted with 3% BSA in PBS overnight at 4 °C. Primary antibodies were detected with 488-conjugated or 594-conjugated secondary antibodies (1:400 dilution; Proteintech #SA00013-7, Proteintech #SA00013-6) for 1 h at room temperature. Finally, after washing three times, the slides were counterstained with H33342 (Sigma). The immunofluorescence signals were detected under the fluorescence microscope (OLYMPUS). The primary antibodies used for immunofluorescence were as follows: rabbit anti-MKX (1:100 dilution; bioss # bs-4468R), rabbit anti-PDCD5 (1:100 dilution; bs-1333R), mouse anti-gamma H2A.X (phospho S139) (1:400 dilution; Abcam ab22551), rabbit anti-Prdm9 (1:200 dilution; Abclonal A20428), mouse anti-IgG (1:400 dilution; Sant Cruzsc sc-515946), and rabbit anti-IgG (1:400 dilution; Abcam ab172730).

## 5. Conclusions

In conclusion, the comprehensive PRDM9 interaction network generated in this study not only enhances our understanding of the molecular basis of DSB formation and repair but also highlights potential new regulatory factors that are involved in spermatogenesis. FUS, MORF4L1, and ARID2 were identified as interacting with PRDM9 to regulate double-strand break repair. In addition to DSB formation and repair, PRDM9 may play a key role in the regulation of alternative RNA splicing and timely degradation of PRDM9, which is essential for meiotic recombination. Future studies will focus on validating these protein interactions in vivo, as we have observed that the expression of the detected genes is not localized in spermatocytes, and the interaction between PRDM9 and other proteins that can be observed in vitro may not occur in vivo.

## Figures and Tables

**Figure 1 ijms-26-01420-f001:**
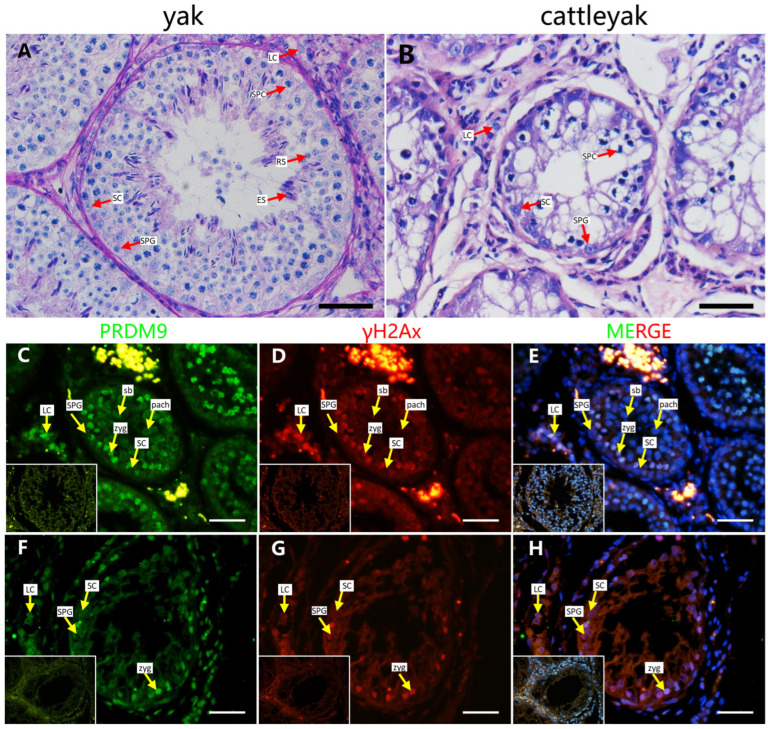
Histomorphology and localization of PRDM9 in the adult yak and cattleyak testes. (**A**) Hematoxylin and eosin (H&E)-stained seminiferous tubule sections showing full spermatogenesis in yak, (**B**) while the cattleyak testis shows spermatogenesis defects. (**C**–**E**) Immunofluorescence staining of PRDM9 (green) and γH2AX (red) in a yak testis. (**F**–**H**) Immunofluorescence staining of PRDM9 (green) and γH2AX (red) in a cattleyak testis (scale bars = 50 μm). The negative control is shown in the bottom left corner of each figure. SPG = spermatogonia; SPCs = spermatocytes; RSs = round spermatids; ESs = elongating spermatids; SCs = Sertoli cells; LCs = Leydig cells; zygs = zygotene spermatocytes; pachs = pachytene spermatocytes; sb = sex body.

**Figure 2 ijms-26-01420-f002:**
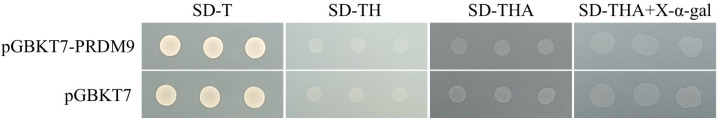
Detection of pGBKT7-PRDM9 self-activation. pGBKT7-PRDM9 and pGBKT7 plasmid-transformed AH109 strains were seeded into an SD-T, SD-TH, SD-THA, SD-THA+X-α-gal nutritional selected medium. T: Trp, TH: Trp/His, THA: Trp/His/Ade.

**Figure 3 ijms-26-01420-f003:**
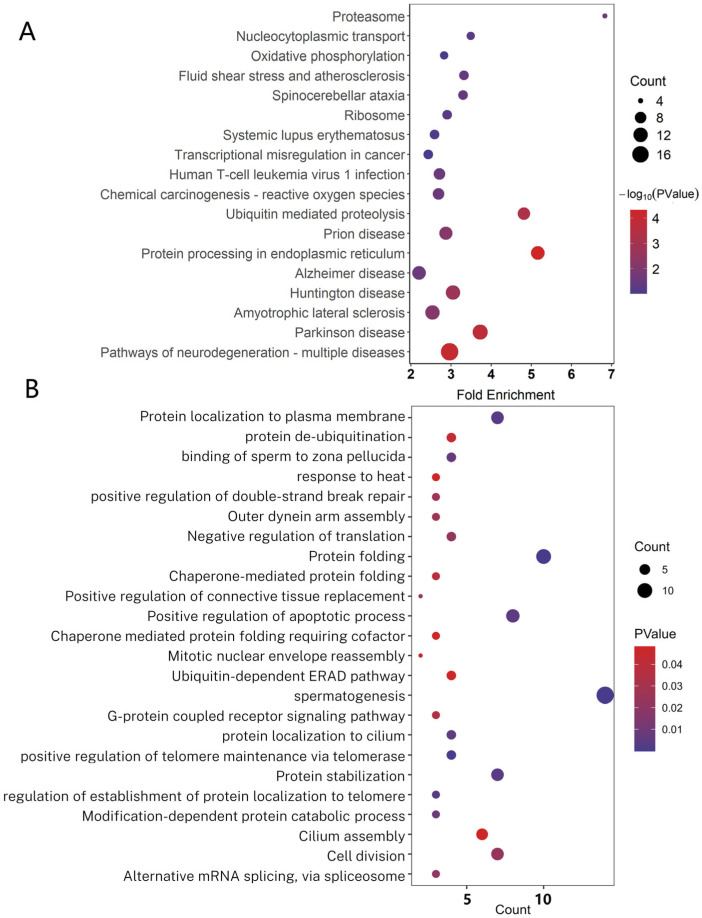
KEGG and GO analysis of PRDM9-interacting genes. (**A**) KEGG analysis of PRDM9-interacting genes. (**B**) GO analysis of PRDM9-interacting genes.

**Figure 4 ijms-26-01420-f004:**
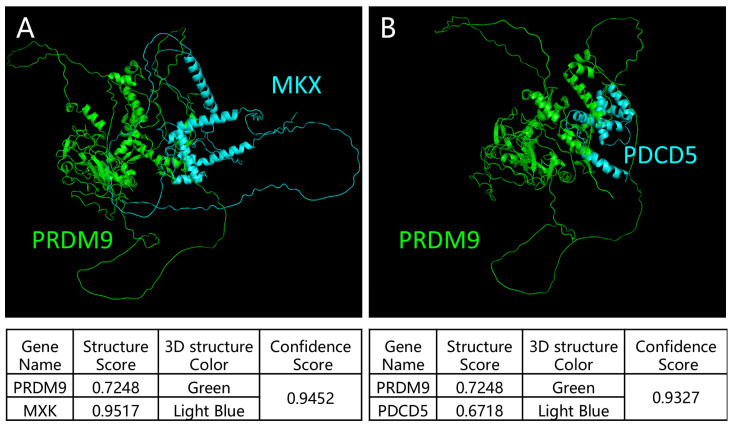
HDOCK server analysis of the interaction between PRDM9 and MKX, as well as PDCD5. (**A**) The crystal structure of the interaction between PRDM9 and MKX. (**B**) The crystal structure of the interaction between PRDM9 and PDCD5.

**Figure 5 ijms-26-01420-f005:**
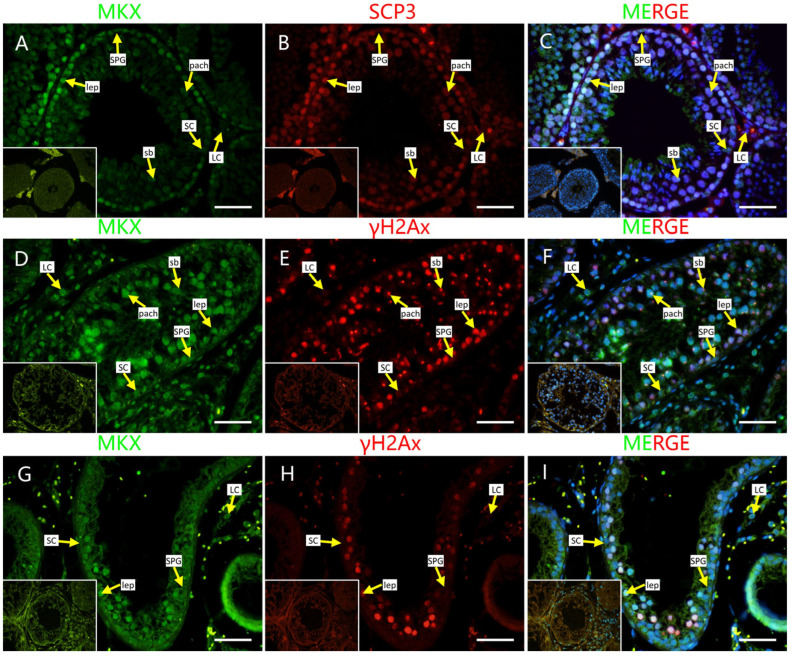
MKX localization in mouse, yak, and cattleyak. (**A**–**C**) MKX localization in adult mouse testis. MKX (green), SCP3 (red). (**D**–**F**) MKX localization in adult yak testis. MKX (green), γH2AX (red). (**G**–**I**) MKX localization in adult cattleyak testis. MKX (green), γH2AX (red) (scale bars = 50 μm). The negative control is shown in the bottom left corner of each figure. SPG = spermatogonia; SCs = Sertoli cells; LCs = Leydig cells; leps = leptene spermtocytes; pachs = pachytene spermatocytes; sb = sex body.

**Figure 6 ijms-26-01420-f006:**
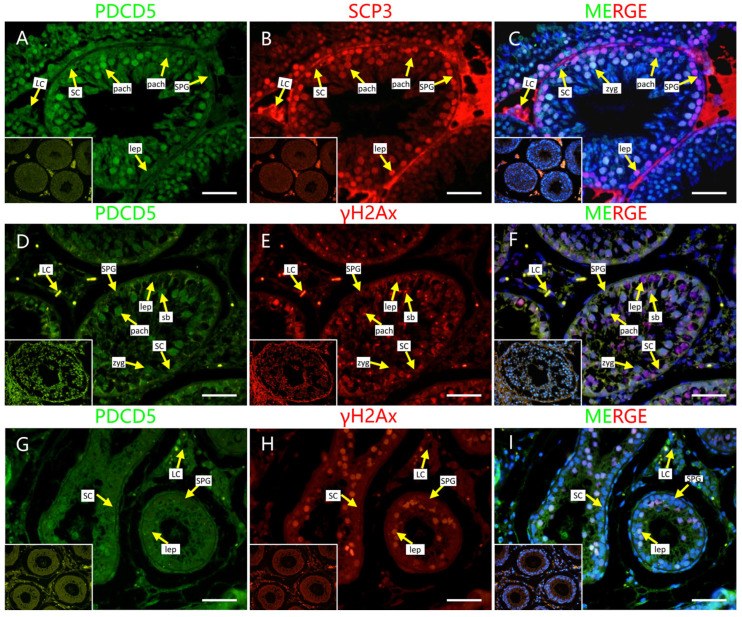
PDCD5 localization in mouse, yak, and cattleyak. (**A**–**C**) PDCD5 localization in adult mouse testis. PDCD5 (green), SCP3 (red). (**D**–**F**) PDCD5 localization in adult yak testis. PDCD5 (green), γH2AX (red). (**G**–**I**) PDCD5 localization in adult cattleyak testis. MKX (green), γH2AX (red) (scale bars = 50 μm). The negative control is shown in the bottom left corner of each figure. SPG = spermatogonia; SCs = Sertoli cells; LCs = Leydig cells; leps = leptene spermtocytes; pachs = pachytene spermatocytes; sb = sex body.

**Figure 7 ijms-26-01420-f007:**
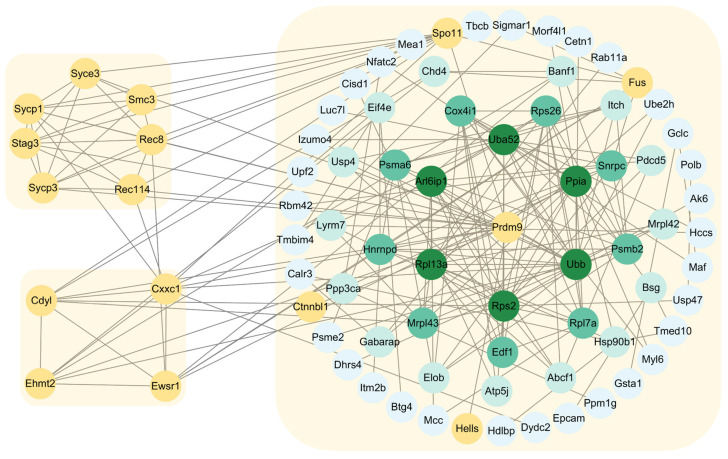
PRDM9 interaction network of yak. The network was created using published mouse data and yak Y2H-seq data from this study. The yellow circles represent proteins that have been proven to directly interact with PRDM9. The others are novel proteins that were found to interact with PRDM9 in yak.

**Table 1 ijms-26-01420-t001:** Annotations of top 20 RDM9-interacting proteins.

Gene Name	Annotation	Reads Counts
PSMA6	Proteasome subunit	67,149
MKX	Homeobox KN domain	52,051
COPS9	Myeloma-overexpressed-like	45,554
SPA17	IQ calmodulin-binding motif	30,014
C3H1orf50	Protein of unknown function (DUF2452)	23,775
TMED10	emp24/gp25L/p24 family/GOLD	23,746
CISD1	Iron-binding zinc finger CDGSH type	23,164
BSG	Immunoglobulin domain	21,796
LOC100297779	GAGE protein	21,367
GSTA1	Glutathione S-transferase, C-terminal domain	21,140
C1H21orf2	Leucine-rich repeat	18,953
ATP5PF	Mitochondrial ATP synthase coupling factor 6	18,702
LOC616478	Proline-rich	18,353
LOC505033	Unknown protein	16,516
PDCD5	Double-stranded DNA-binding domain	14,782
RAB11A	50S ribosome-binding GTPase	14,435
NAGS	NAT, N-acetyltransferase	12,480
LOC100271685	Unknown protein	12,351
TDRD1	Domain of unknown function (DUF4537)	10,740
MORN5	MORN repeat	10,696

## Data Availability

All relevant data discussed in this study are available in the paper and its Appendix A.

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
