# Peer review of "Characterization of PRDM9 Multifunctionality in Yak Testes Through Protein Interaction Mapping"

_ijms, 2025, doi:10.3390/ijms26041420_

Round 1
Reviewer 1 Report
Comments and Suggestions for Authors
Manuscript ijms-3396920
“Title Characterization of PRDM9 Multifunctionality in Yak Testis Through Protein Interaction Mapping” by Wang G. et al.
General comments:
The aim of this manuscript was tried to evaluate the PRDM9 function on spermatogenic cells between yak and cattle-yak for understanding the mechanism of hybrid male sterility. Basically this topic might be interesting for yak-cattle spermatogenesis if there are appropriate experiments and data analysis. However, the present form of the manuscript has problem for publication. Especially, the presentation way for immunohistochemistry. The main problem is the present way is difficult to know whether the specific immunostainings at specific stage of spermatogenic cells are detected or not. If the stage of cells or the cell types are different as indicated in the results, the conclusion will be different. The reviewer strongly recommends to the authors to do additional data to modify the immunofluorescence figures and following the suggestions.
Major comments:
1. Authors should add the negative staining of immunohistochemistry using appropriate IgG at same concentration of 1st antibody or antigen absorbed antibody (Fig. 1, 5, 6). Otherwise, it is difficult to say specific staining of each antibody. Especially, Leydig cells show the positive staining? (Fig. 1, 5, 6)
2. Authors should indicate reinstall clear figures or to add the magnified figures about immunostaining, because present figures are difficult to follow the immunostaining of spematogenic cells at specific stage. (Fig. 1, 5, 6)
3. Authors should add the indication of cells in each figures, such as spg (spermatogonia), lep (leptotene spermatocyte), zyg (zygotene spermatocyte), pach (pachytene spermatocyte), dip (diplotene spermatocyte), rs (round spermatocyte), sp (sperm), sb (sex body), SC (Sertoli cells), LC (Leydig cells), not only in immunofluorescence, but also in HE. (Fig. 1, 5, 6).
4. Depending on the season, spermatogenesis condition in the testis is different because yak and yak-cattle show the seasonal reproduction. Authors should indicate the month of sample collections on yak and cattle-yak. Furthermore, authors should describe the spermatogenesis condition on each sample by testicular cell or tissue examination in the text.
5. PDCD5 was expressed in nuclear of cytoplasm? It is usually translocated from cytoplasm to nuclear when apoptosis occurred. Furthermore, are there so many apoptosis induced spermatocytes in mouse testis? Authors should discuss about it.
6. Furthermore, about the spermatogenic cells apoptosis and proliferation, authors should refer following report (Shimazaki M et al., Reprod Domest Anim 2022) and discuss about the localization of PDCD5.
7. In conclusion, ERAD pathway was described, however, it was not seen in the results and discussion.
Minor comment:
1. Please change gH2Ax into γH2Ax in the figures
Author Response
Comments 1: Authors should add the negative staining of immunohistochemistry using appropriate IgG at same concentration of 1st antibody or antigen absorbed antibody (Fig. 1, 5, 6). Otherwise, it is difficult to say specific staining of each antibody. Especially, Leydig cells show the positive staining? (Fig. 1, 5, 6) |
Response 1:Thank you for pointing this out. We had added the IgG control results in the Fig.1, 5, 6, and pointed the cell types of the germ cells. |
Comments 2: Authors should indicate reinstall clear figures or to add the magnified figures about immunostaining, because present figures are difficult to follow the immunostaining of spematogenic cells at specific stage. (Fig. 1, 5, 6) |
Response 2: Thank you for pointing this out. We have made adjustments to the figures, and added the indication. |
Comments 3: Authors should add the indication of cells in each figures, such as spg (spermatogonia), lep (leptotene spermatocyte), zyg (zygotene spermatocyte), pach (pachytene spermatocyte), dip (diplotene spermatocyte), rs (round spermatocyte), sp (sperm), sb (sex body), SC (Sertoli cells), LC (Leydig cells), not only in immunofluorescence, but also in HE. (Fig. 1, 5, 6). |
Response 3: Thank you for pointing this out. To make the figure understandable for the reader, we have pointed out the cell types in all the figures. |
Comments 4: Depending on the season, spermatogenesis condition in the testis is different because yak and yak-cattle show the seasonal reproduction. Authors should indicate the month of sample collections on yak and cattle-yak. Furthermore, authors should describe the spermatogenesis condition on each sample by testicular cell or tissue examination in the text. |
Response 4: Thank you for pointing this out. We had detected morphology of yak testis in different seasons, however no defects were found in yak spermatogenesis. In this research, all the samples were collected in July. The time of sample collection was added in the text. The description of yak and cattleyak spermatogenesis is added in the text as well. |
Comments 5: PDCD5 was expressed in nuclear of cytoplasm? It is usually translocated from cytoplasm to nuclear when apoptosis occurred. Furthermore, are there so many apoptosis induced spermatocytes in mouse testis? Authors should discuss about it. |
Response 5: Thank you for pointing this out. We had discussed the questions in the text. |
PDCD5 function in multiple cellular processes including apoptosis and autophagy, it facilitates p53-dependent apoptosis through its interaction with mouse double minute 2 homolog (MDM2). It is the first time that PDCD5 expression in mouse spermatocytes has been reported. During meiotic recombination in spermatogenesis, programmed DSBs are formed. P53 is responsible for DNA damage repair, and PDCD5 might be involved in this process. |
Comments 6: Furthermore, about the spermatogenic cells apoptosis and proliferation, authors should refer following report (Shimazaki M et al., Reprod Domest Anim 2022) and discuss about the localization of PDCD5. Response 6: Thank you for pointing this out. We had discussed the questions in the text. In yak and cattleyak, PDCD5 expression started from zygotene spermatocytes and ended at the pachytene stage. In cattleyak, the expression of PDCD5 seems to be lower compared to yak. A previous report showed that the number of apoptotic cells is significantly higher in cattleyak than in yak, which suggests that other pathways might be involved in the apoptosis of germ cells in cattleyak. Comments 7: In conclusion, ERAD pathway was described, however, it was not seen in the results and discussion. Response 7: Thank you for pointing this out. In this study, according to the GO analysis, the interacting proteins were enriched in the ubiquitin-dependent ERAD pathway. However, no further data support the conclusion. What we mean is that the degradation of PRDM9 might depend on the ubiquitin-dependent ERAD pathway. Therefore, we have removed this point in the conclusion. Comments 8: Please change gH2Ax into γH2Ax in the figures Response 8: Thank you for pointing this out. We had modified the description. |
Reviewer 2 Report
Comments and Suggestions for Authors
Meiotic recombination is initiated by the formation of programmed DNA double-strand breaks during spermatogenesis. PRDM9 determines the localization of recombination hotspots by interacting with several protein complexes, but the function of PRDM9 in spermatogenesis of mice or yaks still largely unknown. In this study, Wang et al., established a PRDM9 interaction network, and novel function of PRDM9, which will provide new insights for understanding spermatogenesis and male sterility. However, some major revisions are required before this paper can be accepted.
Major comments:
1. In the introduction section, authors should add more details about spermatogenesis of mammalian.
2. In Figure 1A, B, it is unclear how spermatogenesis is disrupted in cattleyak, more details should be provided in the figure.
3. I suggest the authors perform KEGG and GSEA analysis on the PRDM9 interacting protein.
4. Co-inmunoprecipitation (CoIP) should be performed to verified the interaction between PRDM9 and PDCD5 or MXK.
5. In discussion section, authors mention AlphaFold and RFdiffusion are also popular for predicting protein-protein interactions. Why not further analysis the interaction between PRDM9 and the target proteins using these methods.
Minor comments:
1. the contents of acknowledgments and appendix are incorrect.
2. the position of Figure 1 is incorrect.
3. There are still some grammatical errors in the article, the author needs to check carefully and revise it.
Comments on the Quality of English LanguageThe quality of English language can be improved.
Author Response
Response 1:Thank you for pointing this out. And in my opinion, it is necessary to add more details about spermatogenesis. The additional information of spermatogenesis is added in the main text. |
Comments 2: In Figure 1A, B, it is unclear how spermatogenesis is disrupted in cattleyak, more details should be provided in the figure. |
Response 2: Thank you for pointing this out. To make the figure understandable for the reader, we have pointed out the cell types in all the figures. |
Comments 3: I suggest the authors perform KEGG and GSEA analysis on the PRDM9 interacting protein. |
Response 3: We agree with this suggestion. Therefore, we have performed the analysis and provided some explanations in the main text. |
Comments 4: Co-inmunoprecipitation (CoIP) should be performed to verified the interaction between PRDM9 and PDCD5 or MXK. |
Response 4: Thank you for pointing this out. Y2H applied in this research is a powerful tool to detect protein interaction, however false positive results still exist. Further experiments are needed to confirm the interaction, such as Co-IP. However, the antibody's applicability does not support Co-IP. For this reason, we applied the HDOCK server, which is widely accepted, to verify the interaction. Of course, all the interacted proteins will be validated in subsequent work. |
Comments 5: In discussion section, authors mention AlphaFold and RFdiffusion are also popular for predicting protein-protein interactions. Why not further analysis the interaction between PRDM9 and the target proteins using these methods. |
Comments 7: the contents of acknowledgments and appendix are incorrect. |
Response 7: Thank you for pointing this out. We had made modifications. |
Comments 8: . the position of Figure 1 is incorrect.. |
Response 8: Thank you for pointing this out. We had made modifications. |
Comments 9: .There are still some grammatical errors in the article, the author needs to check carefully and revise it. |
Response 9: Thank you for pointing this out. We had check the grammatical errors and made modifications. |
Round 2
Reviewer 1 Report
Comments and Suggestions for Authors
The authors have corrected many areas according to the comments, but some areas are still not clear due to lack of explanation or appropriate photographs. Also, scale bars are needed in the inset figures (Figs. 1, 5, and 6). Since the molecular analysis was performed on testicular tissues, the various type of cells in the testis should also be noted to represent the antibody staining results. The authors should carefully explain the results in the text and the figures should be clearly presented. Furthermore, authors should modify the discussion where the parts are related to the following changing the results.
For example, Fig.1C.D.E, the strong staining in the upper center and lower right is not seen in negative controls, and those may be Leydig cell staining, which needs to be explained in the text. Those strong staining in the stroma was not seen in cattle-yak (Fig.1F,G,H), and seems different from Yak, which may need to add explain in the manuscript.
About Fig.5 A, B, C mouse tissues showed the strong staining in stroma of negative control, but not in the antibody staining. Reviewer thinks if the same concentration of negative control showed strong staining in stroma, then the antibody should show the same strong staining of stroma. Authors should explain about them. About Fig. 5G, H. I, quite strong staining was observed at the left side center and the right side center to bottom. Authors should explain or reinstall appropriate figures.
About Fig.5 A, B, C mouse tissues showed the strong staining in stroma in SCP3 and negative control, then authors should describe in the manuscript, they are not specific staining at Leydig cells. About Fig.5D negative controls seems positive staining of spermatogenic cells and Leydig cells. Is it negative control? Furthermore, Cytoplasm of Sertoli cells seems many PDCD5 staining (see at the bottom of the seminiferous tubules, 4 Sertoli cells were observed).
For the negative control, authors should use rabbit IgG for rabbit polyclonal antibody and mouse IgG for mouse monoclonal antibody. However, authors describe only mouse IgG.
Author Response
Dear Reviewer ,
Thank you so much for giving us reasonable suggestions to improve the quality of the manuscript. To my understanding, your main concern is the immunostaining of the testis, which is important for the presentation of the results. The staining has been done several times, and the results are the same. The details of the response to the comments are as follows:
Comments 1: The authors have corrected many areas according to the comments, but some areas are still not clear due to lack of explanation or appropriate photographs. Also, scale bars are needed in the inset figures (Figs. 1, 5, and 6). Since the molecular analysis was performed on testicular tissues, the various type of cells in the testis should also be noted to represent the antibody staining results. The authors should carefully explain the results in the text and the figures should be clearly presented. Furthermore, authors should modify the discussion where the parts are related to the following changing the results.
Response 1:
I think the scale bars are inserted in all the staining images at the bottom right corner, and all the images, including the negative control at the bottom left corner, are at the same magnification. I’m not sure whether the scale bars are needed in the negative control, as they seem to be too short to be recognizable.
The antibodies γH2Ax and SCP3 are putative markers used to label spermatocytes. The cells types are recognized according to the nuclear type and markers. We have made modifications to the results and the corresponding part in the discussion as well.
Comments 2: For example, Fig.1C.D.E, the strong staining in the upper center and lower right is not seen in negative controls, and those may be Leydig cell staining, which needs to be explained in the text. Those strong staining in the stroma was not seen in cattle-yak (Fig.1F,G,H), and seems different from Yak, which may need to add explain in the manuscript.
Response 2:
During sample preparation, the blood cells in the blood vessels could not be washed clean. The strong staing in the Fig.1C.D.E are the autofluorescence of blood cells. The interstitial cells might also exhibit the autofluorescence phenomenon.
The localization of γH2Ax mainly expressed in nuclear of cells. PRDM9 functions in the localization of DSB hotspots and combines with DNA. Because of the difference in cell types between yak and cattleyak, they do show different expression. However, PRDM9 and γH2Ax are localized in the leptotene and zygotene spermatocytes in both yak and cattleyak.
Comments 3: About Fig.5 A, B, C mouse tissues showed the strong staining in stroma of negative control, but not in the antibody staining. Reviewer thinks if the same concentration of negative control showed strong staining in stroma, then the antibody should show the same strong staining of stroma. Authors should explain about them. About Fig. 5G, H. I, quite strong staining was observed at the left side center and the right side center to bottom. Authors should explain or reinstall appropriate figures.
Response 3:
About Fig.5 A, B, C, the interstitial cells were strongly stained in negative control. As mentioned above, interstitial cells also exhibit the autofluorescence phenomenon. In both the green and red channels, the interstitial cells are strongly stained, which demonstrates that the strong staining is caused by autofluorescence.
About Fig. 5G, H. I , the tissue detachment from the slides led to non-specific staining at the right side center to bottom. At the left side center, the blood vessels and blood cells showed autofluorescence. We replaced Fig. 5G, H. I.
Comments 4: About Fig.5 A, B, C mouse tissues showed the strong staining in stroma in SCP3 and negative control, then authors should describe in the manuscript, they are not specific staining at Leydig cells. About Fig.5D negative controls seems positive staining of spermatogenic cells and Leydig cells. Is it negative control? Furthermore, Cytoplasm of Sertoli cells seems many PDCD5 staining (see at the bottom of the seminiferous tubules, 4 Sertoli cells were observed).
Response 4:
We made description of the non-specific staining in interstitial cell in the main text. About Fig.5D, the positive signal observed in the negative control is non-specific staining. All the positive signals do not localized in nuclears.
PDCD5 was also expressed in Sertoli cells, in the main text, we had made description.
Comments 5: For the negative control, authors should use rabbit IgG for rabbit polyclonal antibody and mouse IgG for mouse monoclonal antibody. However, authors describe only mouse IgG.
Response 5:
We applied corresponding IgG of the staining, the information of the IgG is added in the method part.
Reviewer 2 Report
Comments and Suggestions for Authors
The authors have improved the manuscript based on my suggestions, I think it it suitable to be accepted
Author Response
Dear Reviewer ,
Thank you so much for giving us reasonable suggestions to improve the quality of the manuscript.
Round 3
Reviewer 1 Report
Comments and Suggestions for Authors
Reviewer thinks authors responded my comments well and revised the manuscript appropriately.
Author Response
Comments 1:Reviewer thinks authors responded my comments well and revised the manuscript appropriately.
Response 1:Thank you for taking the time to review this manuscript. All the comments provided are reasonable and will greatly help improve the quality of the work.
